# Shifting temporal trends and disparities in sarcoidosis mortality in the United States: A retrospective analysis from 1999 to 2020

**Fatima Ali Raza[1], Sumeet Kumar[2], Ayesha Mohammad[3], Areej Amin[4], Farooq Ahmad[5], Sohaib Tousif[6], Urwah Kamran[5], Lamea Bint Sahab[7], Sajjad Ali[6], Mah I. Kan Changez[8]\*, Aman Goyal[9]\***

1 Department of Medicine, Karachi Medical and Dental College, Karachi, Pakistan, 2 Department of Medicine, Dow University of Health Science, Karachi, Pakistan, 3 Department of Medicine, Comanche County Memorial Hospital, Lawton, Oklahoma, United States of America, 4 Department of Medicine, Rawal institute of Health Sciences, Islamabad, Pakistan, 5 Department of Medicine, Allama Iqbal Medical College, Lahore, Pakistan, 6 Department of Medicine, Ziauddin Medical University, Karachi, Pakistan, 7 Department of Medicine, Jinnah Sindh Medical University, Karachi, Pakistan, 8 Department of Cardiothoracic Surgery, Yale University, New Haven, Connecticut, United States of America, 9 Department of Internal Medicine, Seth GS Medical College and KEM Hospital, Mumbai, India

\* mahikan.changez@yale.edu (MIKC); amanmgy@gmail.com (AG)

**Data Availability Statement:** All relevant data are within the manuscript and its Supporting Information files.

## Abstract

### Introduction

Sarcoidosis is an inflammatory disease characterized by granulomas, the etiology of which remains unclear. This study examines sarcoidosis-related mortality trends in the United States from 1999 to 2020, with a focus on disparities pertaining to patient sex, geographical location, and urbanization status.

### Methods

We analyzed death certificate data from the CDC WONDER database, using ICD-10 code D86. Age-adjusted mortality rates (AAMR) per 1,000,000 people were calculated. Trends were analyzed using Joinpoint regression models to determine annual percentage changes (APC).

### Results

A total of 37,956 Sarcoidosis-related deaths were documented from 1999 to 2020 in the United States. The AAMR increased from 3.9 in 1999 to 6.4 in 2020. Significant mortality increases were observed from 1999–2001 and again from 2018–2020. Mortality rates were consistently higher among women compared to men. A significant difference in AAMR was observed across states, with highest mortality in the South region and lowest in the West region. Urbanization trends shifted from higher AAMR rates in metropolitan to non-metropolitan areas post-2018. Non-Hispanic Black individuals experienced the highest mortality rates throughout the study period.

**Funding:** The author(s) received no specific funding for this work.

**Competing interests:** The authors have declared that no competing interests exist.

**Abbreviations:** 1. AAMR, Age-Adjusted Mortality Rate; 2. APC, Annual Percentage Change; 3.CDC, Centers for Disease Control and Prevention; 4. ICD-10, International Classification of Diseases, 10th Revision; 5. STROBE, Strengthening the Reporting of Observational Studies in Epidemiology; 6. CI, Confidence Interval.

## Conclusions

This study highlights significant racial and geographic disparities in sarcoidosis-related mortality. Women, Black patients, and those residing in non-metropolitan areas are at the highest risk for Sarcoidosis associated mortality. Targeted public health interventions are required to address these prevalent disparities.

## Introduction

Sarcoidosis is a systemic inflammatory disease characterized by the formation of granulomas —microscopic clusters of immune cells—most commonly affecting the lungs and lymphatic system [1]. Although the precise etiology remains unknown, it is hypothesized that sarcoidosis results from an exaggerated immune response to an unidentified antigen, with a disproportionately higher incidence observed among Black populations [2, 3].

Previous studies have documented sarcoidosis-related mortality trends in the United States from 1999 to 2016, highlighting notable demographic and geographic disparities [4]. However, an updated analysis incorporating more recent developments is warranted and remains unexplored. We hypothesize that the plateaued mortality rate of sarcoidosis observed in their study might have experienced an inflection point since then, with a potential change in mortality based on urbanization status.

This study extends the evaluation of sarcoidosis mortality trends through 2020, with a specific focus on recent changes and disparities based on urbanization status. This study aims to provide new insights into the impact of living environments on sarcoidosis outcomes and inform targeted public health interventions to mitigate the disease burden among vulnerable populations.

## Methods

### Study setting and population

We conducted a descriptive analysis of death certificate data from the Centers for Disease Control and Prevention's Wide-Ranging Online Data for Epidemiologic Research (WONDER) database, spanning the years 1999 to 2020.

This analysis concentrated on death certificates within the Multiple Cause of Death Public Use dataset. We identified cases where sarcoidosis was listed as a primary or contributing cause of death, using the International Classification of Diseases, 10th Revision (ICD-10) code D86.

Institutional Review Board approval was not required, as we utilized a de-identified public dataset. This study adheres to the Strengthening the Reporting of Observational Studies in Epidemiology (STROBE) guidelines.

### Data abstraction

Data were categorized by demographics, including, sex, and location of death (medical facilities, nursing or long-term care facilities, hospices, homes or other unclassified sites). Geographic categorization was based on census regions and states as defined by the United States Census Bureau. Urbanization levels were classified using the National Center for Health Statistics Urban Rural Classification Scheme, distinguishing metropolitan areas from non-metropolitan areas.

## Data synthesis

Sarcoidosis mortality trends in the United States from 1999 to 2020 were analyzed, considering sex, state, urbanization, and census regions. Crude and age-adjusted mortality rates (AAMR) per 1000,000 people were calculated, using the 2000 census for AAMR standardization.

Piecewise log-linear regression models were employed to detect trend changes using the Joinpoint Regression Program (Version 5.2.0, National Cancer Institute) [5]. Age-adjusted mortality variations were reported as annual percent change (APC) with 95% confidence intervals. Statistical significance was assessed with a two-tailed t-test with a threshold of P<0.05.

For data indicating zero to nine (0–9) deaths, the rates were suppressed. When there are less than ten people in the population, the corresponding denominator population estimates are likewise suppressed. When there were fewer than 20 deaths, rates were labeled as "unreliable." Rates with "not stated" or unknown age or ethnicity were denoted as "not applicable" when the denominator population number was not available. AAMRs were not computed for deaths of individuals whose ages were "not stated" or unknown.

## Ethics statement

Institutional Review Board approval and patient consent were not required for our study, as all data were retrospectively analyzed from a publicly accessible, de-identified, and anonymized database. The data were accessed from the CDC Wonder Database on 10/15/2024.

## Results

A total of 37956 Sarcoidosis-related deaths were documented in the United States between 1999 and 2020 (**S1 Table**). Of these, 63.8% occurred within medical facilities, 24.9% occurred in home, 7.62% occurred in nursing home/long-term care facilities, and 3.5% occurred at hospice (**S2 Table**).

## Annual trends for sarcoidosis-related mortality

The AAMR for Sarcoidosis-related deaths in the United States was 3.9 in 1999, rising roughly to double 6.4 in 2020 (**S3 Table**). The AAMR showed a significant increase from 1999 to 2001, with an APC of 9.66 (95% CI, 2.26 to 15.92). From 2001 to 2018, the trend plateaued with a minimal increase in AAMR, reflected by an APC of 0.52 (95% CI, -0.65 to 0.77). However, an increase in mortality was observed from 2018 to 2020, with an APC of 8.54 (95% CI, 2.66 to 11.52) (**S4 Table**).

## Sarcoidosis-related AAMR stratified by sex

Women consistently had higher age-adjusted sarcoidosis mortality rates than men, with rates increasing from 4.4 per 100,000 in 1999 to 6.6 per 100,000 in 2020 (**S3 Table**).

Initially, from 1999 to 2001, there was a significant rise with an APC of 13.43%, followed by a slight decline between 2001 and 2018 (APC: -0.45%). However, from 2018 to 2020, mortality rates for women surged again with an APC of 9.20% (**S4 Table**).

In contrast, men started with a lower mortality rate of 3.2 per 100,000 in 1999, which increased to 6.1 per 100,000 by 2020 (**S3 Table**). From 1999 to 2018, the increase in men's mortality was moderate (APC: 2.01%) but became more pronounced from 2018 to 2020, with an APC of 7.43% (**S4 Table**, **Fig 1**).

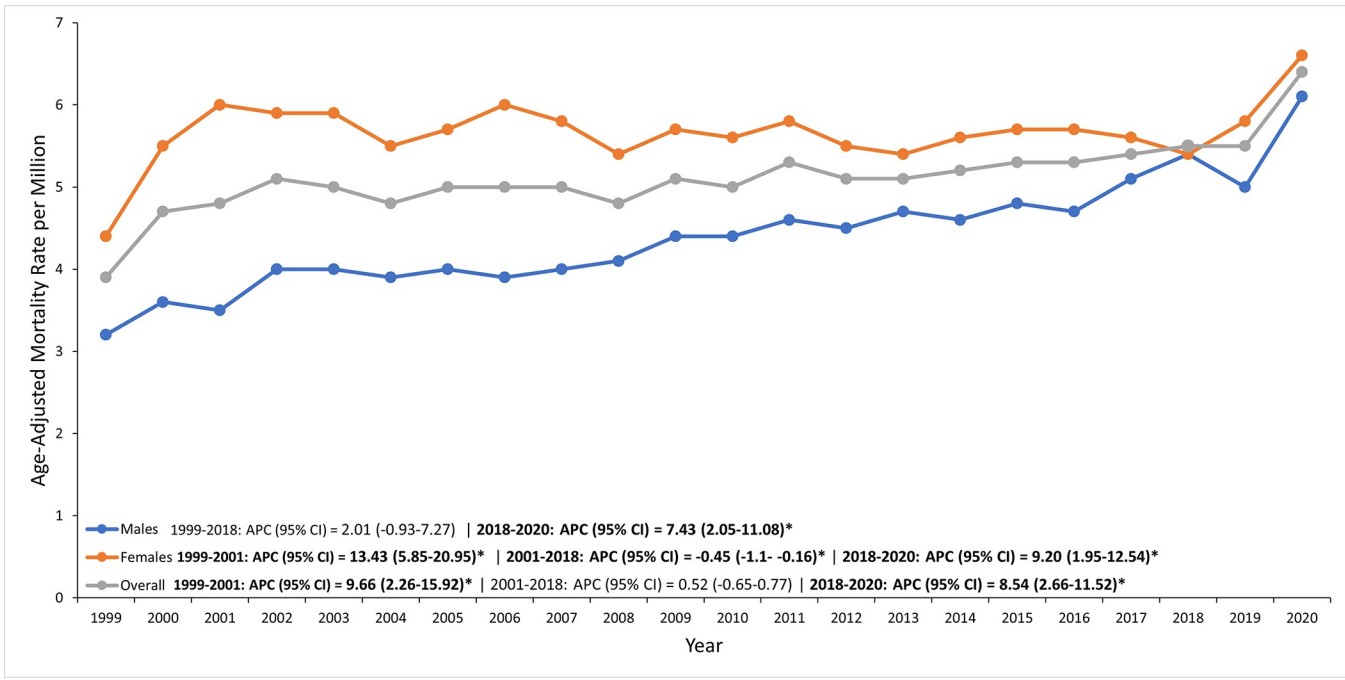

**Fig 1. Sex-based disparities in sarcoidosis-related AAMRs in the U.S. from 1999 to 2020, with women experiencing higher rates than men.**

## Sarcoidosis-related AAMR stratified by state and census region

A significant difference in AAMR was observed across states, with the 2020 AAMRs ranging from the lowest AAMR of 0.7 per 100,000 [95% CI: 0.4–1.0] in Hawaii to the highest AAMR of 21.9 per 100,000 [95% CI: 19.3–24.4] in the District of Columbia (**Fig 2**).

States that fell into the top 90th percentile, including the District of Columbia, South Carolina, and Maryland, were compared with states in the lower 10th percentile, namely Hawaii, Arizona, and New Mexico (**S5 Table**).

States with the highest overall AAMR change from 2003 to 2020 were observed in the District of Columbia, South Carolina, and North Carolina, reflecting significant increases over the study period.

On average, over the course of the study period, the highest mortality was observed in the South region (AAMR: 6.1 per 1,000,000, 95% CI: 5.6–6.5), followed by the Northeast region (AAMR: 5.3 per 1,000,000, 95% CI: 4.8–5.8), the Midwest region (AAMR: 4.8 per 1,000,000, 95% CI: 4.3–5.3), and the West region (AAMR: 3.3 per 1,000,000, 95% CI: 2.9–3.7) (**S6 Table**, **Fig 3**).

## Sarcoidosis-related AAMR stratified by urbanization status

From 1999 to 2018, metropolitan areas consistently had higher Sarcoidosis-related AAMRs compared to non-metropolitan areas. Following 2018, this trend began to shift, with non-metropolitan areas showing increasingly higher AAMRs than metropolitan areas (**S7 Table**, **Fig 4**).

The overall APC for metropolitan areas was [APC: 0.39% (95% CI: -3.93–0.66)], indicating a generally stable trend with minor fluctuations. In contrast, the APC for non-metropolitan areas was [APC: 1.34% (95% CI: -0.65–2.21)], suggesting a modest upward trend over the

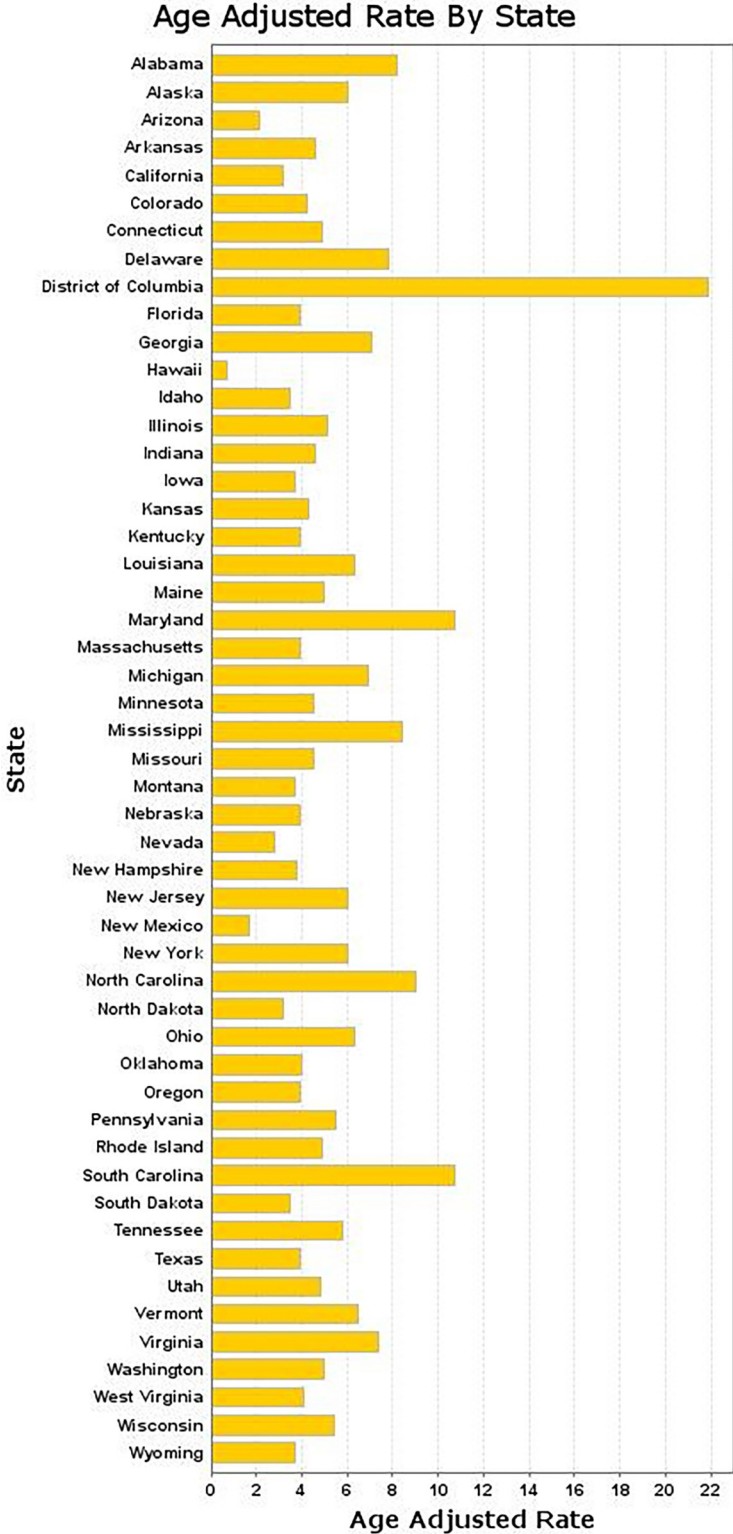

**Fig 2. Bar chart comparing sarcoidosis-related AAMRs across U.S. states, highlighting geographic disparities in mortality.**

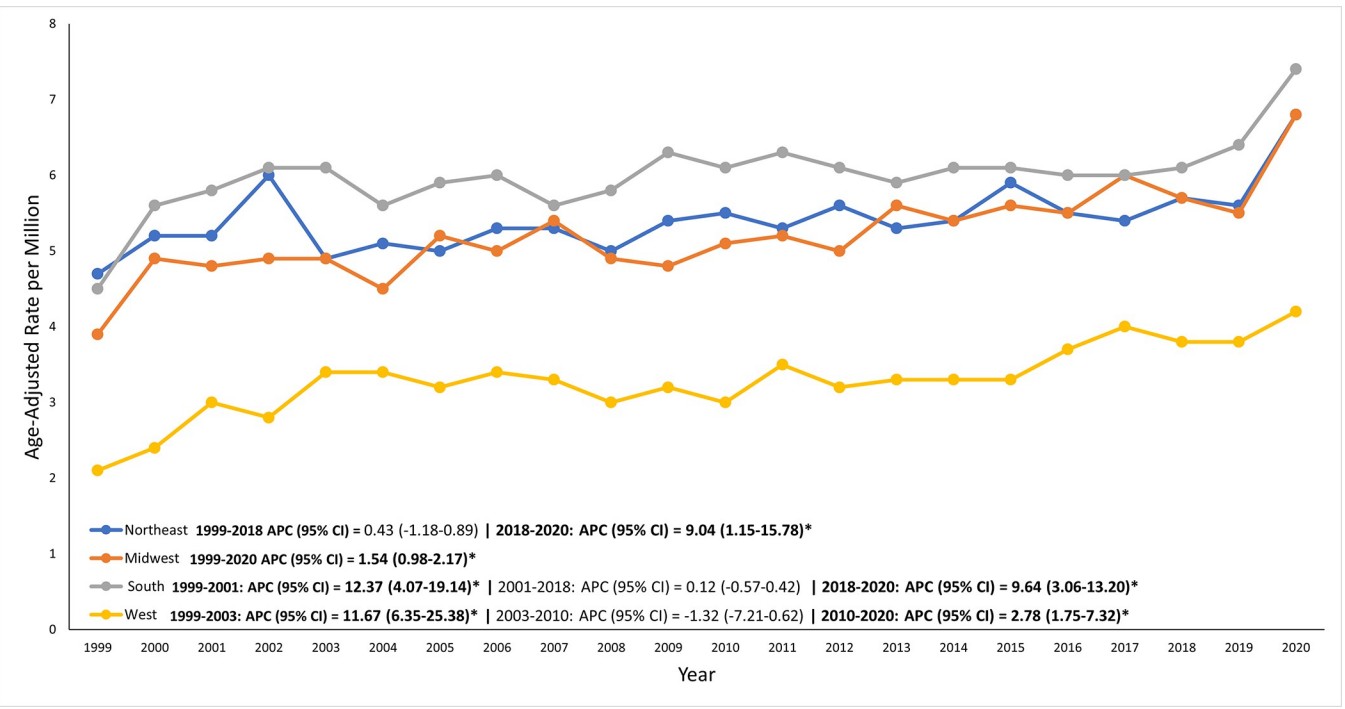

**Fig 3. Regional differences in sarcoidosis-related AAMRs across U.S. Census regions, with notable disparities seen among different census regions.**

entire study period (**S4 Table**). indicating the significant rise in non-metropolitan areas in recent years.

### Sarcoidosis-related AAMR stratified by race

Throughout the study period, Non-Hispanic Black individuals consistently experienced the highest AAMRs for Sarcoidosis, peaking at 27.4 per 100,000 in 2002 and 28.4 per 100,000 in 2020. Despite some fluctuations, their AAMRs remained significantly higher than those of other racial and ethnic groups. In contrast, Non-Hispanic White individuals had much lower AAMRs but showed a steady increase from 1.9 per 100,000 in 1999 to 4.1 per 100,000 in 2020 (**Fig 5**).

For Hispanic individuals, AAMRs were generally low, fluctuating between 1.2 and 2.1 per 100,000 over the years, with a gradual rise from 1.2 in 1999 to 2.1 in 2020. The data for Non-Hispanic American Indian or Alaska Native and Non-Hispanic Asian or Pacific Islander populations were often unreliable due to small sample sizes, but when available, the AAMRs were relatively low and variable, indicating less consistent trends (**S8 Table**).

### Discussion

This study highlights significant racial and geographic disparities in sarcoidosis AAMRs in the United States from 1999 to 2020. Black patients consistently experienced the highest AAMRs, aligning with previous studies [6, 7]. This disparity may be influenced by factors such as socioeconomic status, genetics, and systemic racial discrimination [8]. Geographic trends also reflect earlier findings, with substantial increases in AAMRs from 2003 to 2020, particularly in the District of Columbia, South Carolina, and North Carolina [6, 7]. Nam et al. similarly

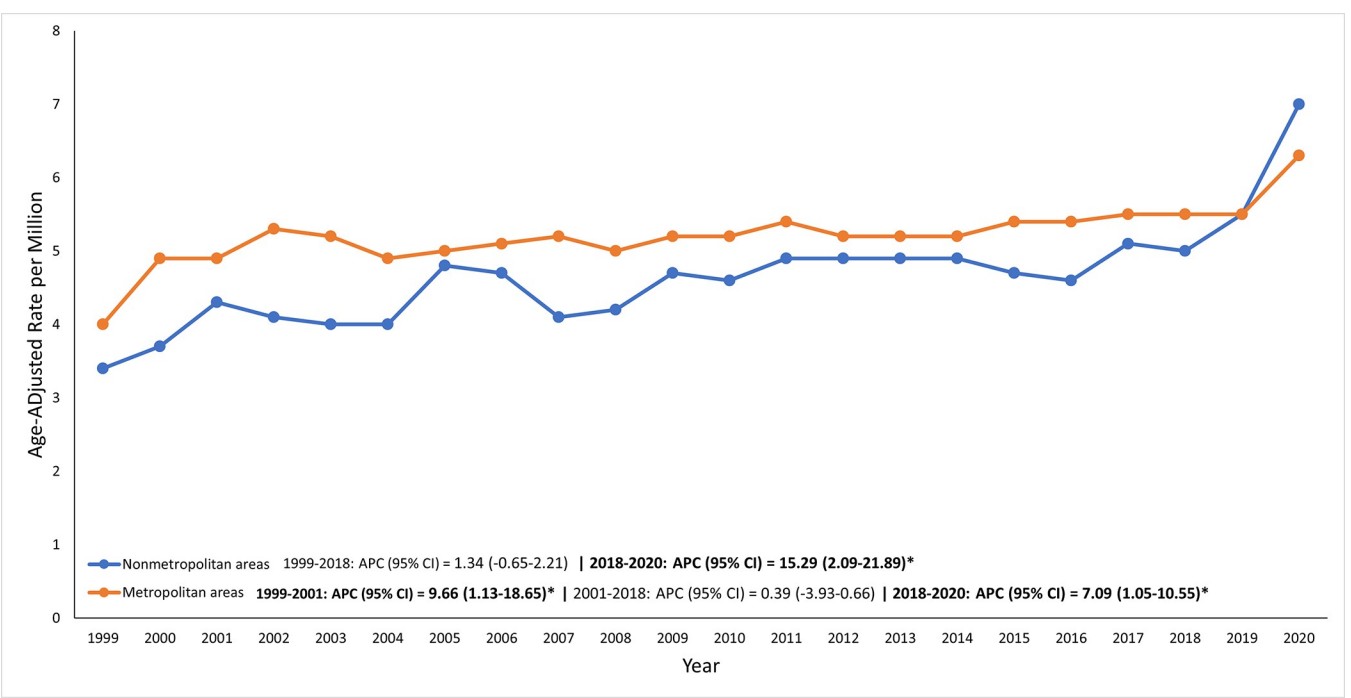

**Fig 4. The impact of urbanization on sarcoidosis-related AAMRs, showing variation between rural and urban settings.**

identified high prevalence in the East Coast, attributed to urbanization, racial composition, and socioeconomic inequalities [9].

An unexpected finding was the shift in higher AAMRs from metropolitan to non-metropolitan areas after 2018. Initially, metropolitan areas had higher mortality rates, consistent with previous studies [9]. However, the rise in non-metropolitan areas may be explained by migration patterns leading to an aging population and higher mortality rates among adults aged 55–64 [10]. Non-metropolitan areas also showed increased poverty rates and prevalence of chronic diseases in 2018 [8, 11], which are linked to higher mortality. Additionally, these regions have seen a rise in racial diversity and limited healthcare access, highlighted by the closure of 138 non-metropolitan hospitals from 2010 to 2021 [11, 12].

The South consistently exhibited the highest AAMRs, from 4.5 per million in 1999 to 7.4 per million in 2020, potentially due to the higher prevalence of sarcoidosis among Black patients in this region [13, 14]. The Northeast also had elevated mortality rates, likely influenced by socioeconomic disparities and environmental exposures [7, 14]. Conversely, the Midwest's modest increase and the West's lower rates may be related to regional agricultural exposures and better healthcare access, respectively [15].

Mortality trends were marked by three phases: an initial sharp increase from 1999 to 2001 (APC 9.66%), likely due to advancements in diagnostic techniques [16]; a stabilization period from 2001 to 2018 (APC 0.52%) reflecting treatment improvements [17, 18]; and a resurgence from 2018 to 2020 (APC 8.54%). One possible reason for this recent inflection point could be variations in the coding practices for sarcoidosis-related mortality by physicians, as well as an increased rate of diagnosis and documentation of mortality facilitated by advanced imaging modalities such as high-resolution computed tomography and positron emission tomography scans [19]. Furthermore, despite advancements in therapeutics, adequate management for advanced fibrocystic cases remains unavailable [20], which may further contribute to our

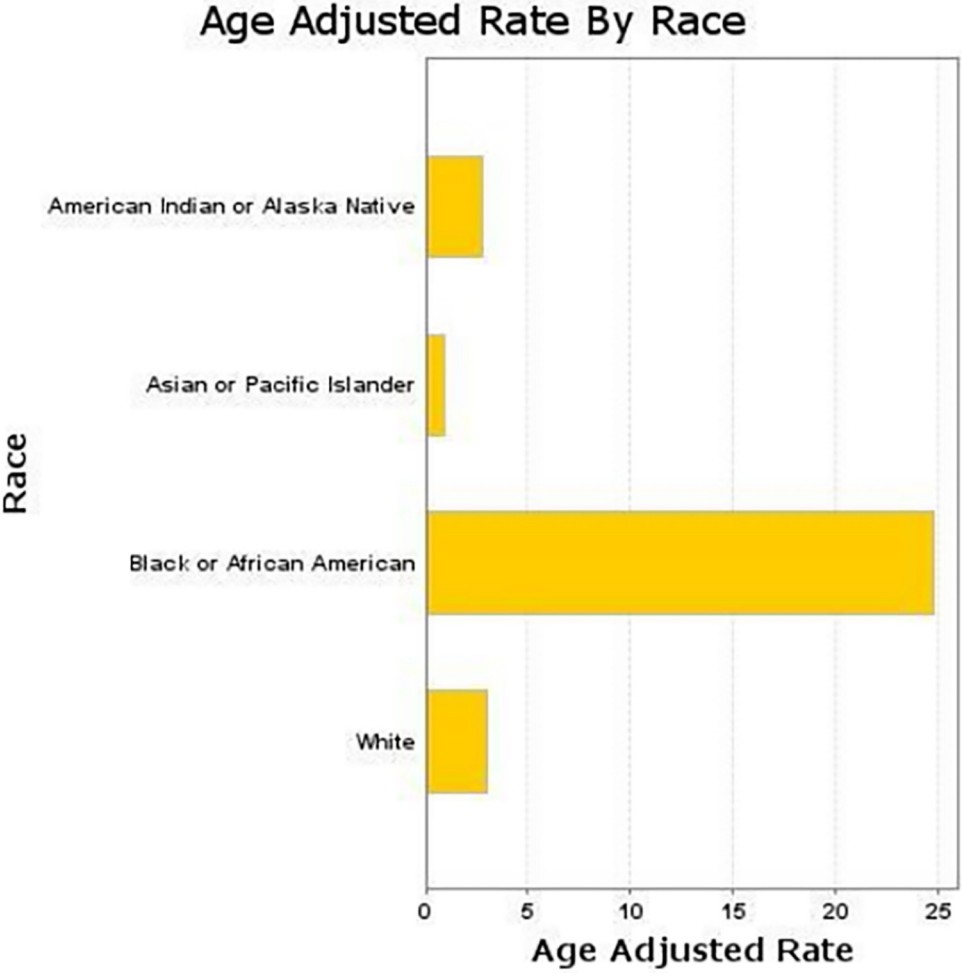

**Fig 5. Racial disparities in sarcoidosis-related AAMRs, with Black individuals experiencing the highest rates.**

findings. Further prospective studies with standard guidelines need to be developed to manage more complex cases of sarcoidosis. Sex disparities were notable, with women generally experiencing higher mortality rates than men, except during the stabilization period, possibly due to hormonal factors [21]. We hypothesize that the sharp increase in mortality among men during 2018–2020 may relate to delayed diagnosis and higher baseline comorbidity rates, and increased vulnerability during the pandemic [22].

To better understand these shifting trends, further research is essential. Cohort studies comparing outcomes between patients with and without COVID-19, along with investigations into healthcare access and treatment continuity during the COVID-19 pandemic, are critical. These efforts will help determine whether the observed mortality increase and the shifting disparities—such as higher mortality in non-metropolitan areas—are partly attributable to the COVID-19 pandemic or other factors. Ultimately, this research will guide targeted interventions to improve outcomes for sarcoidosis patients.

## Limitations

There are several limitations to this study that should be considered. First, the reliance on ICD codes and death certificates for sarcoidosis-related mortality introduces the potential for

misclassification or omission, as sarcoidosis may not always be listed as a primary cause of death, particularly in cases where complications or comorbidities were more prominent. Second, variations in clinical practices and diagnostic criteria for sarcoidosis over the study period may have affected the consistency and comparability of mortality data across different regions and time points. Third, this study did not have access to data on medical therapies or treatment regimens for sarcoidosis, which could have provided insights into the impact of evolving treatment practices on mortality outcomes. Lastly, socioeconomic factors, such as income, education, and health insurance status, which are critical determinants of healthcare access and outcomes, were not available in the dataset. This limitation makes it challenging to fully understand how disparities in access to care may have contributed to regional variations in mortality associated with sarcoidosis.

## Conclusion

This study provides a detailed examination of sarcoidosis-related mortality in the U.S. from 1999 to 2020, revealing significant differences based on race, geography, and time. Black patients consistently had the highest AAMRs, likely due to a mix of socioeconomic challenges, genetic factors, and unequal access to healthcare. Geographically, the Southern states showed the highest death rates, which may be related to the larger Black population and differences in healthcare availability and environmental factors in the region.

The study identified three key phases in mortality trends: an initial sharp rise from 1999 to 2001, a period of stability from 2001 to 2018, and a resurgence from 2018 to 2020. Moreover, the trends in mortality rates began shifting from urban to rural areas after 2018, likely due to reduced access to specialized care and the closure of healthcare facilities in non-metropolitan regions. Women had higher mortality rates than men.

These findings highlight the urgent need for public health efforts to address these disparities.

## Supporting information

**S1 Table. Sarcoidosis related deaths, stratified by sex, in the United States, 1999 to 2020.**
(DOCX)

**S2 Table. Sarcoidosis related mortality, stratified by place of death in the United States, 1999 to 2020.**
(DOCX)

**S3 Table. Overall and sex-stratified sarcoidosis related age-adjusted mortality rates per 1,000,000 in the United States, 1999 to 2020.**
(DOCX)

**S4 Table. Annual percent change (APC) of sarcoidosis related age-adjusted mortality rates per 1,000,000 in the United States, 1999 to 2020, 1999 to 2020.**
(DOCX)

**S5 Table. State-stratified sarcoidosis related age-adjusted mortality rates per 1,000,000 in the United States, 1999 to 2020.**
(DOCX)

**S6 Table. Census region-stratified sarcoidosis related age-adjusted mortality rates per 1,000,000 in the United States, 1999 to 2020.**
(DOCX)

**S7 Table. Urbanization-stratified sarcoidosis related age-adjusted mortality rates per 1,000,000 in the United States, 1999 to 2020.**
(DOCX)

**S8 Table. State-stratified sarcoidosis related age-adjusted mortality rates per 1,000,000 in the United States, 1999 to 2020.**
(DOCX)

## Author Contributions

**Conceptualization:** Fatima Ali Raza, Aman Goyal.

**Data curation:** Fatima Ali Raza, Sumeet Kumar, Urwah Kamran.

**Formal analysis:** Ayesha Mohammad, Areej Amin, Farooq Ahmad, Urwah Kamran, Sajjad Ali, Aman Goyal.

**Funding acquisition:** Sumeet Kumar.

**Investigation:** Fatima Ali Raza, Ayesha Mohammad, Areej Amin, Farooq Ahmad, Sohaib Tousif, Urwah Kamran, Lamea Bint Sahab.

**Methodology:** Sumeet Kumar, Ayesha Mohammad, Areej Amin, Farooq Ahmad, Sohaib Tousif, Lamea Bint Sahab.

**Project administration:** Ayesha Mohammad, Sajjad Ali.

**Resources:** Sumeet Kumar, Areej Amin, Sohaib Tousif, Urwah Kamran.

**Software:** Sumeet Kumar, Ayesha Mohammad, Farooq Ahmad, Sohaib Tousif, Urwah Kamran, Lamea Bint Sahab, Sajjad Ali.

**Supervision:** Fatima Ali Raza, Sajjad Ali, Aman Goyal.

**Validation:** Fatima Ali Raza, Aman Goyal.

**Visualization:** Sohaib Tousif, Sajjad Ali, Aman Goyal.

**Writing – original draft:** Fatima Ali Raza, Sumeet Kumar, Ayesha Mohammad, Areej Amin, Farooq Ahmad, Sohaib Tousif, Urwah Kamran, Lamea Bint Sahab, Mah I. Kan Changez, Aman Goyal.

**Writing – review & editing:** Sajjad Ali, Mah I. Kan Changez, Aman Goyal.

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
