## [Decision Letter · Decision Letter 0]

1 Dec 2024

PONE-D-24-50326Disparities in Mortality from Sarcoidosis by Age, Gender, Ethnoracial Background, and Housing Status: A Retrospective Analysis from 1999 to 2020PLOS ONE

Dear Dr. Changez,

Thank you for submitting your manuscript to PLOS ONE. After careful consideration, we feel that it has merit but does not fully meet PLOS ONE’s publication criteria as it currently stands. Therefore, we invite you to submit a revised version of the manuscript that addresses the points raised during the review process.

We look forward to receiving your revised manuscript.

Kind regards,

Amr Ehab El-Qushayri

Academic Editor

PLOS ONE

2. We note that Figure 3 in your submission contain [map/satellite] images which may be copyrighted. All PLOS content is published under the Creative Commons Attribution License (CC BY 4.0), which means that the manuscript, images, and Supporting Information files will be freely available online, and any third party is permitted to access, download, copy, distribute, and use these materials in any way, even commercially, with proper attribution. For these reasons, we cannot publish previously copyrighted maps or satellite images created using proprietary data, such as Google software (Google Maps, Street View, and Earth). For more information, see our copyright guidelines: http://journals.plos.org/plosone/s/licenses-and-copyright.

a. You may seek permission from the original copyright holder of Figure 3 to publish the content specifically under the CC BY 4.0 license. 

Additional Editor Comments:

I would like to congratulate the authors for this good paper.

I have some comments to be addressed by the authors:

1-You included 2020 in your study and reflected it on COVID-19 pandemic. What is the last date of the recruitment period in 2020? As WHO declared the pandemic in March 2020.

2-If you have the cause of death, it would be worth noting to report it to exclude if it is COVID-19 related or not, as well as, to know the trends of causes of death. If you can not access it, it is not recommended to report that it is related to COVID-19 directly or indirectly.

Reviewers' comments:

Reviewer's Responses to Questions

**Comments to the Author**

1. Is the manuscript technically sound, and do the data support the conclusions?

Reviewer #1: Yes

Reviewer #2: Yes

2. Has the statistical analysis been performed appropriately and rigorously? 

Reviewer #1: Yes

Reviewer #2: Yes

3. Have the authors made all data underlying the findings in their manuscript fully available?

Reviewer #1: Yes

Reviewer #2: Yes

4. Is the manuscript presented in an intelligible fashion and written in standard English?

Reviewer #1: Yes

Reviewer #2: Yes

5. Review Comments to the Author

Reviewer #1: The manuscript is well-structured and well-analyzed and provides paramount results. The figures are greatly depicted and are structured and clear. The Discussion section is comprehensive and offers a detailed analysis. Furthermore, the study's limitations are effectively addressed.

General comments:

Strengths:

1:

Conclusion part is appropriately detailed and comprehensive. And provides a full overview of the whole article.

2:

Additionally, limitations of the study are provided in a separate and dedicated section, which is highly informative.

Minor revision points:

3:

Clarity of Figures 1-6, while informative, could benefit from more detailed explanations and consistent formatting to improve readability.

4:

It is advised to incorporate an additional section following the conclusion to clearly define and list all abbreviations used in the article.

Thanks to the authors for their hard work on this Study. It’s a pleasure to review such a contribution to the field.

Reviewer #2: Overall it's a well written paper and easy to understand. There is not much comment from me except to suggest to put all the figures after the description text. eg. figure 1 should be following the text for sarcoidosis related AAMR stratified by gender.

6. PLOS authors have the option to publish the peer review history of their article (what does this mean?). If published, this will include your full peer review and any attached files.

Reviewer #1: **Yes: **SEYED AMIRHOSSEIN MAZHARI

Reviewer #2: No

---

## [Author Response · Author response to Decision Letter 0]

9 Dec 2024

Dear Respectful Editor and Reviewers,

Thank you for your feedback on our manuscript titled “Shifting Temporal Trends and Disparities in Sarcoidosis Mortality in the United States: A Retrospective Analysis from 1999 to 2020.” We appreciate your recognition of the study's strengths and the valuable insights you provided.

Below we are attaching our in-depth response to each query by the reviewers:

Editor’s Comments:

I would like to congratulate the authors for this good paper.

I have some comments to be addressed by the authors:

1-You included 2020 in your study and reflected it on COVID-19 pandemic. What is the last date of the recruitment period in 2020? As WHO declared the pandemic in March 2020.

2-If you have the cause of death, it would be worth noting to report it to exclude if it is COVID-19 related or not, as well as, to know the trends of causes of death. If you can not access it, it is not recommended to report that it is related to COVID-19 directly or indirectly.

-> Dear Editor,

The CDC Wonder database includes data up to the end of each calendar year, with 2020 data extending through the last day of December. However, as per your appropriate suggestion in point 2, the CDC Wonder database does not provide information on causes of death. Accordingly, we have removed any direct or indirect mentions of COVID-19 as a cause, as well as references to it in any hypothesis statements. We would be happy to make any further changes you suggest.

We have added this to our discussion, instead of the COVID-19 hypothesis:

“One possible reason for this recent inflection point could be variations in the coding practices for sarcoidosis-related mortality by physicians, as well as an increased rate of diagnosis and documentation of mortality facilitated by advanced imaging modalities such as high-resolution computed tomography and positron emission tomography scans [19]. Furthermore, despite advancements in therapeutics, adequate management for advanced fibrocystic cases remains unavailable [20], which may further contribute to our findings. Further prospective studies with standard guidelines need to be developed to manage more complex cases of sarcoidosis. Sex disparities were notable, with women generally experiencing higher mortality rates than men, except during the stabilization period, possibly due to hormonal factors [21]. We hypothesize that the sharp increase in mortality among men during 2018-2020 may relate to delayed diagnosis and higher baseline comorbidity rates, and increased vulnerability during the pandemic [22]. 

To better understand these shifting trends, further research is essential. Cohort studies comparing outcomes between patients with and without COVID-19, along with investigations into healthcare access and treatment continuity during the COVID-19 pandemic, are critical. These efforts will help determine whether the observed mortality increase and the shifting disparities—such as higher mortality in non-metropolitan areas—are partly attributable to the COVID-19 pandemic or other factors. Ultimately, this research will guide targeted interventions to improve outcomes for sarcoidosis patients.”

2. Review Comments to the Author

2.1. Reviewer #1: 

The manuscript is well-structured and well-analyzed and provides paramount results. The figures are greatly depicted and are structured and clear. The Discussion section is comprehensive and offers a detailed analysis. Furthermore, the study's limitations are effectively addressed.  Strengths:  1: Conclusion part is appropriately detailed and comprehensive. And provides a full overview of the whole article.  2: Additionally, limitations of the study are provided in a separate and dedicated section, which is highly informative.  Minor revision points:  3: Clarity of Figures 1-6, while informative, could benefit from more detailed explanations and consistent formatting to improve readability.

-> Dear reviewer, as per your suggestion, we have now added more detail into the figure caption for each figure in the Figure Legend section of our manuscript after the References.   4: It is advised to incorporate an additional section following the conclusion to clearly define and list all abbreviations used in the article.

-> Dear reviewer, a new subheader after the conclusion section for Abbreviations has been created as per your suggestion.   Thanks to the authors for their hard work on this Study. It’s a pleasure to review such a contribution to the field.

-> Dear Reviewer, thank you for your thoughtful comments regarding the strength and novelty of our manuscript.

2.2. Reviewer #2: 

Overall it's a well written paper and easy to understand. There is not much comment from me except to suggest to put all the figures after the description text. eg. figure 1 should be following the text for sarcoidosis related AAMR stratified by gender.

-> Dear reviewer, thank you for your suggestions. We have added all figures separately as TIFF file in the editorial manager as per the journal’s requirement. However, we have provided the figure caption right after the text, and an additional figure legend at end of the manuscript as per your suggestion.

Thank you,

Mah I Kan Changez, MBBS

Yale University

Mahikan.changez@yale.edu

---

## [Decision Letter · Decision Letter 1]

22 Dec 2024

PONE-D-24-50326R1Shifting Temporal Trends and Disparities in Sarcoidosis Mortality in the United States: A Retrospective Analysis from 1999 to 2020PLOS ONE

Dear Dr. Changez,

Thank you for submitting your manuscript to PLOS ONE. After careful consideration, we feel that it has merit but does not fully meet PLOS ONE’s publication criteria as it currently stands. Therefore, we invite you to submit a revised version of the manuscript that addresses the points raised during the review process.

We look forward to receiving your revised manuscript.

Kind regards,

Amr Ehab El-Qushayri

Academic Editor

PLOS ONE

Journal Requirements:

Reviewers' comments:

Reviewer's Responses to Questions

**Comments to the Author**

Reviewer #1: All comments have been addressed

Reviewer #3: (No Response)

2. Is the manuscript technically sound, and do the data support the conclusions?

Reviewer #1: Yes

Reviewer #3: Yes

3. Has the statistical analysis been performed appropriately and rigorously? 

Reviewer #1: Yes

Reviewer #3: Yes

4. Have the authors made all data underlying the findings in their manuscript fully available?

Reviewer #1: Yes

Reviewer #3: Yes

5. Is the manuscript presented in an intelligible fashion and written in standard English?

Reviewer #1: Yes

Reviewer #3: Yes

6. Review Comments to the Author

Reviewer #1: Some Minor revisions points have been suggested and they all have been addressed

appropriately.

Minor revision points:

Thanks to the authors for their hard work on this Revision. It’s a pleasure to review such a

contribution to the field.

Reviewer #3: I appreciate your response to the editor’s suggestion regarding the deletion of content related to COVID-19 mortality. However, I noticed that this content is still present in the abstract and keywords. I recommend that you replace it in these sections as you did in the manuscript and remove it from the keywords, as it is not central to the core of the research. Additionally, on page 8, you used the abbreviation “AAPC.” If this was intended to be “APC,” please correct it. If not, kindly provide an explanation for “AAPC” in the abbreviation list. Thank you for your efforts.

7. PLOS authors have the option to publish the peer review history of their article (what does this mean?). If published, this will include your full peer review and any attached files.

Reviewer #1: **Yes: **SEYED AMIRHOSSEIN MAZHARI

Reviewer #3: No

---

## [Author Response · Author response to Decision Letter 1]

23 Dec 2024

Dear Respectful Editor and Reviewers,

Thank you for your feedback on our manuscript titled “Shifting Temporal Trends and Disparities in Sarcoidosis Mortality in the United States: A Retrospective Analysis from 1999 to 2020.” We appreciate your recognition of the study's strengths and the valuable insights you provided.

Below we are attaching our in-depth response to each query by the reviewers:

Reviewer #1: Some Minor revisions points have been suggested and they all have been addressed

appropriately.

Minor revision points:

Thanks to the authors for their hard work on this Revision. It’s a pleasure to review such a

contribution to the field.

-> Dear Reviewer, Thank you for your suggestions and for your feedback that the work is ready for publication.

Reviewer #3: I appreciate your response to the editor’s suggestion regarding the deletion of content related to COVID-19 mortality. However, I noticed that this content is still present in the abstract and keywords. I recommend that you replace it in these sections as you did in the manuscript and remove it from the keywords, as it is not central to the core of the research. Additionally, on page 8, you used the abbreviation “AAPC.” If this was intended to be “APC,” please correct it. If not, kindly provide an explanation for “AAPC” in the abbreviation list. Thank you for your efforts.

-> Dear Reviewer,

We have updated the Abstract and removed the COVID-19 section to reflect the changes made during the Round 1 revisions, which we had inadvertently omitted from the submission portal. Additionally, we have revised the keywords and changed “AAPC” to “APC” in accordance with your comments. Thank you for your valuable feedback.

Please let us know if you’d like any further changes.

Thank you,

Mah I Kan Changez, MBBS

Yale University

Mahikan.changez@yale.edu

---

## [Editor Report · Decision Letter 2]

26 Dec 2024

Shifting Temporal Trends and Disparities in Sarcoidosis Mortality in the United States: A Retrospective Analysis from 1999 to 2020

PONE-D-24-50326R2

Dear Dr. Mah I Kan Changez,

We’re pleased to inform you that your manuscript has been judged scientifically suitable for publication and will be formally accepted for publication once it meets all outstanding technical requirements.

Kind regards,

Amr Ehab El-Qushayri

Academic Editor

PLOS ONE

---

## [Editor Report · Acceptance letter]

31 Dec 2024

PONE-D-24-50326R2 

PLOS ONE

Dear Dr. Changez, 

I'm pleased to inform you that your manuscript has been deemed suitable for publication in PLOS ONE. Congratulations! Your manuscript is now being handed over to our production team.

Kind regards, 

on behalf of

Dr. Amr Ehab El-Qushayri 

Academic Editor

PLOS ONE